# Effects of Vitamin D Deficiency on Sepsis

**DOI:** 10.3390/nu15204309

**Published:** 2023-10-10

**Authors:** Hyeri Seok, Jooyun Kim, Won Suk Choi, Dae Won Park

**Affiliations:** Division of Infectious Diseases, Department of Medicine, Korea University Ansan Hospital, Korea University College of Medicine, Ansan 15355, Republic of Korea; hyeri.seok@gmail.com (H.S.);

**Keywords:** vitamin D, vitamin D deficiency, sepsis, septic shock, mortality

## Abstract

A prospective cohort study was conducted to evaluate the effect of vitamin D deficiency on sepsis. A total of 129 patients were enrolled. The median age was 74 years old, with a median SOFA score of 7; septic shock was observed in 60 patients. The median vitamin D level in the overall population was 13 ng/mL. A total of 96 patients had vitamin D deficiency, whereas 62 patients were described to have severe vitamin D deficiency. Severe vitamin D deficiency significantly increased the 14-day mortality (adjusted hazard ratio (aHR) 2.57; 95% confidence interval [CI]: 1.03–6.43; *p* = 0.043), 28-day mortality (aHR 2.28; 95% CI: 1.17–4.45; *p* = 0.016), and in-hospital mortality (aHR 2.11; 95% CI: 1.02–4.36; *p* = 0.044). In Kaplan–Meier analysis, the severe vitamin D deficiency group had significantly higher 14-day and 28-day mortality rates compared with the non-deficient group. Evaluating the vitamin D levels in sepsis patients may become necessary in an aging society. Severe vitamin D deficiency can independently affect poor prognosis related to sepsis. Further studies are needed to evaluate whether vitamin D supplementation in sepsis patients with vitamin D deficiency can help improve the prognosis of sepsis in addition to improving bone mineral metabolism.

## 1. Introduction

Sepsis refers to a syndrome related to physiological, pathological, and biochemical abnormalities that are caused by the host’s systemic immune response to infections; both innate and acquired immunity are involved in the host’s immune response [1,2]. Sepsis is a major cause of morbidity and mortality worldwide and results in increased hospital costs and hospitalization days [3]. The definition and bundle approach for the management of sepsis was introduced more than 30 years ago in 1991. Since that time, the death rate from sepsis has decreased from its original 60–70% mortality rate, although is still remains high, with a worldwide rate of 20% and a rate of 30% in Korea [3,4,5]. Although the American Society of Critical Care and the European Society of Critical Care revised the third international standard definition and international guidelines for sepsis and sepsis shock in 2016, the definition of sepsis and its management remains controversial. Moreover, treatments that significantly improve the mortality rate of sepsis are limited to early antibiotic use, norepinephrine use, and low tidal volumes on a mechanical ventilator [1,6].

The proportion of elderly patients suffering from sepsis has increased as the average life expectancy increases. According to previous studies, sepsis patients had a median age of 72 years old and people over 65 years of age accounted for 75% of sepsis patients [5]. The independent factors associated with high mortality in sepsis patients were being elderly and suffering from chronic diseases [7]. Patients with comorbidity or residing in long-term facilities were independent causal-related factors that increased the 14-day and 28-day mortality rates in sepsis patients, respectively. Since the elderly, individuals with chronic diseases, and long-term facility residents are not correctable variables, remediable approaches are needed for these high-risk groups for sepsis.

The U.S. National Health and Nutrition Examination Survey (NHANES) reported that the percentage of individuals with serum vitamin D deficiency increased to 36% during 2001–2004, compared with 22% between 1988 and 1994 [8,9]. According to data from the Health Insurance Review and Assessment Service, vitamin D deficiency in Korea increased from about 3000 in 2010 to about 31,000 in 2014, providing an average annual growth rate of 7.9%, which is higher than the Organization for Economic Cooperation and Development (OECD) average (8.9%) [10]. The main route of vitamin D supply is its synthesis through sunlight exposure, whereas factors that affect this ability include the season, old age, obesity, and chronic diseases; moreover, individuals that insufficiently synthesize vitamin D are recommended to administer vitamin D supplements [11]. According to a previous study in Korea, 77.9% of all elderly patients in the study group suffered from vitamin D deficiency and the level of vitamin D deficiency was eight times higher in elderly residents living in long-term care facilities than in the community [12]. The group at high risk of being vitamin D deficient and the group at high risk of sepsis morbidity and mortality have some common factors; thus, it is necessary to evaluate the effectiveness of vitamin D in the common high-risk group.

The main role of vitamin D is in mineral and bone metabolism, which is related to the development of metabolic bone diseases, including osteoporosis [11]. Recent studies have focused on the role of vitamin D in cell proliferation and differentiation in muscles and the immune system and found an association between vitamin D deficiency and cancer, infectious diseases, and autoimmune diseases, as well as cardiovascular diseases and diabetes [13,14]. Vitamin D can modulate innate and adaptive immune responses by regulating T helper cell differentiation and proliferation and promoting tolerogenic immune responses [15]. The role of vitamin D in the immune system begins through the vitamin D receptor (VDR) that is present in activated T cells, B cells, neutrophils, macrophages, and dendritic cells [16]. In terms of innate immunity, vitamin D induces the expression of cathelicidins and beta-defensin [17]. In infectious diseases, the incidences of pneumonia and influenza virus infections decreased in randomized controlled trials following vitamin D supplementation [18,19]. The effect of vitamin D deficiency on sepsis is controversial. Previous prospective studies and meta-analyses have reported that vitamin D deficiency was associated with increased mortalities in sepsis patients [20,21,22]. However, vitamin D deficiency was not associated with mortality from sepsis in further randomized controlled studies [23,24], and no study has shown that vitamin D supplementation reduces the mortality rate of sepsis. This study was conducted to evaluate the relationship between vitamin D deficiency and mortality in sepsis patients after the introduction of the Sepsis-3 definition.

## 2. Materials and Methods

### 2.1. Study Design, Study Population, and Data Collection

A prospective cohort study was conducted to evaluate the association between vitamin D deficiency and the prognosis of sepsis. The study was conducted among adult patients who were older than 18 years with a qSOFA score of 2 or higher who had visited the emergency room at the Korea University Ansan Hospital between January 2019 and January 2020 and who agreed to participate in the study. According to the Sepsis-3 definition, patients with a SOFA score of 2 were defined as having sepsis, whereas septic shock was defined in patients where a vasopressor was required to maintain a minimum mean arterial pressure of 65 mmHg and a serum lactate level of at least 2 mmol/L.

The following clinical data were collected for patients with sepsis. The demographic data, such as age and sex, the underlying disease, and the Charlson comorbidity index (CCI), were collected. SOFA scores were collected in addition to the blood test results for C-reactive protein (CRP), procalcitonin, and lactate at the initial sepsis diagnosis. The timing at which the antibiotics were first administered following the sepsis diagnosis and the appropriateness of the initial empirical antibiotics were also collected. As outcome factors, 7-day, 14-day, 28-day, and in-hospital mortalities were collected in addition to the length of hospitalization. This study protocol was approved by the institutional review board of the Korea University Ansan Hospital (no. 2020AS0345). Written informed consent was obtained from each study participant or legal representative if not available.

### 2.2. Definition

The vitamin D measured in this study was 25-hydroxy vitamin D (25-OH vitamin D), which was quantitatively measured via chemiluminescent microparticle immunoassay with serum samples. Vitamin D was measured using a test method available in a typical laboratory, where a qualified standard laboratory undergoes regular investigations; therefore we waived intraassay and interassay factors. Vitamin D deficiency was defined as a vitamin D level below 20 ng/mL, whereas severe vitamin D deficiency was defined as a level below 12 ng/mL. Vitamin D insufficiency was defined as greater than 20 ng/mL and less than 30 ng/mL.

### 2.3. Statistical Analysis

For comparison, Pearson χ^2^ tests and Fisher’s exact tests were used for categorical variables and Student’s *t*-test, Mann–Whitney U tests, and Kruskal–Wallis tests were used for continuous variables, as appropriate. Cox proportional hazard regression analysis was used to evaluate the association between vitamin D deficiency and mortality. Variables adjusted in the Cox regression model included age, sex, Charlson comorbidity index, SOFA score, CRP levels, procalcitonin levels, lactate levels, and the timing and appropriateness of the antibiotic treatments. All statistical analyses were performed using SPSS Statistics version 20.0 for Windows (IBM Corp., Armonk, NY, USA).

## 3. Results

During the study period, a total of 129 patients were enrolled. The median vitamin D level in the overall population was 13 ng/mL (interquartile range: 6.2–20.2). A total of 96 patients (74.4%) were vitamin D deficient, 62 patients (48.0%) had severe vitamin D deficiency, and 119 patients (92.2%) had vitamin D insufficiency. The baseline characteristics of the study population are presented in Table 1. In the vitamin D deficiency group, 42 (43.8%) participants were male, whereas 27 (43.5%) males were in the severe vitamin D deficiency group. High-grade infections occurred in 60 (62.5%) and 36 (37.5%) participants in the vitamin D deficiency and severe vitamin D deficiency groups, respectively. The median SOFA score for vitamin D deficiency, severe vitamin D deficiency, and vitamin D insufficiency group was 7 ng/mL. Bacteremia was observed in 36 (37.9%) individuals with vitamin D deficiency and in 21 (33.9%) with severe vitamin D deficiency. A total of 46 (47.9%) patients had septic shock in the vitamin D deficient group, whereas 29 (46.8%) patients had septic shock who were severely vitamin D deficient. The number of patients admitted to an intensive care unit (ICU) was 63 (65.6%) and 39 (62.9%) in the vitamin D deficient and severe vitamin D deficient groups, respectively. The median C-reactive protein levels were 10.2 mg/dL and 11.2 mg/dL in the vitamin D deficient and severe vitamin D deficient groups, respectively, and the median procalcitonin levels were 1.08 ng/mL and 0.90 ng/mL in vitamin D deficient and severe vitamin D deficient groups, respectively. The initial median lactate level was 2.7 mmol/L in the vitamin D deficient group and 2.3 mmol/L in the severe vitamin D deficient group. The median time for starting antibiotics was 115 and 130 min, and the appropriateness of antibiotics was 75.0% and 80.6% in the vitamin D deficient and severe vitamin D deficient groups, respectively. The all-cause 7-day mortality was 10.1% in the overall population, 11.6% in the vitamin D deficiency group, 14.5% in the severe vitamin D deficiency, and 10.9% in the vitamin D insufficiency group. The all-cause 14-day mortality was 17.1% in the overall population, 20.8% in the vitamin D deficiency, 24.6% in the severe vitamin D deficiency, and 18.5% in the vitamin D insufficiency group. The all-cause 28-day mortality was 29.5% in the overall population, 32.3% in the vitamin D deficiency, 38.7% in the severe vitamin D deficiency, and 30.3% in the vitamin D insufficiency group. The in-hospital mortality rate was 27.9% in the overall population, 31.3% in the vitamin D deficiency, 35.5% in the severe vitamin D deficiency, and 28.6% in the vitamin D insufficiency group.

The results of the variables that affect the sepsis mortality rate, including vitamin D deficiency, are presented in Table 2. The mortality rates were evaluated as 7-day mortality, 14-day mortality, 28-day mortality, and in-hospital mortality, respectively. SOFA was significantly increased in the 7-day mortality sepsis patients in the multivariate analysis (adjusted hazard ratio (aHR): 1.22; 95% confidence interval [CI]: 1.02–1.44; *p*-value = 0.025). Charlson comorbidity index was significantly increased in the 14-day mortality sepsis patients (aHR: 1.30; 95% CI: 1.11–1.53; *p* = 0.002). SOFA (aHR: 1.22; 95% CI: 1.07–1.39; *p* = 0.003) and Charlson comorbidity index (aHR: 1.11; 95% CI: 1.01–1.22; *p* = 0.027) statistically increased in the 28-day mortality patients. Charlson comorbidity index significantly increased in-hospital mortality (HR: 1.31; 95% CI: 1.14–1.50; *p* < 0.001). Vitamin D deficiency did not affect mortality in any of the sepsis patients.

Table 3 shows the results of analyzing the variables associated with severe vitamin D deficiency that affect the sepsis mortality rate. Although vitamin D deficiency did not affect sepsis mortality, severe vitamin D deficiency significantly increased the 14-day mortality (aHR: 2.57; 95% CI: 1.03–6.43; *p* = 0.043), 28-day mortality (aHR: 2.28; 95% CI: 1.17–4.45; *p* = 0.016), and in-hospital mortality (aHR: 2.11; 95% CI: 1.02–4.36; *p* = 0.044). The results of SOFA increasing the 7-day and 28-day mortality and the Charlson comorbidity index increasing the 14-day, 28-day, and in-hospital morbidity remained unchanged. Lactate, shock, and ICU admission did not significantly affect any mortality rates when performing the univariate analysis (*p* > 0.01).

The results of the Kaplan–Meier analysis on the effects of vitamin D deficiency and severe vitamin D deficiency on the mortality rates of sepsis are shown in Figure 1 and Figure 2. The vitamin D deficiency group showed a lower tendency than the non-deficient group in terms of 7-day, 14-day, and 28-day mortalities, although not to any level of significance. In the Kaplan–Meier analysis of the severe vitamin D deficiency group and the non-severe group, there were statistically significant differences in the 14-day and 28-day mortalities. The 14-day mortality rates in the severe vitamin D deficiency group and the non-severe group were 10.4% and 24.6%, respectively (*p* = 0.048). The 28-day mortality rate in the severe vitamin D deficiency group and the non-severe group were 20.9% and 39.3%, respectively (*p* = 0.026). In Kaplan–Meier analysis, vitamin D insufficiency had no statistically significant effect on the mortality rates of sepsis.

## 4. Discussion

This study analyzed the effects of vitamin D deficiency on the mortality rates in sepsis and sepsis shock patients. To our knowledge, this is the first study conducted in sepsis patients since the introduction of Sepsis-3 definition. Severe vitamin D deficiency increased the 14-day, 28-day, and in-hospital mortality rates in sepsis patients, demonstrated by the higher adjusted hazard ratios than the SOFA and Charlson comorbidity index values, although vitamin D deficiency did not affect the mortality rate.

The pathophysiology whereby vitamin D deficiency affects the mortality rate from sepsis is not yet clear. Previous studies reported that vitamin D deficiency was associated with a decrease in the concentration of vitamin D receptors (VDR) and that the level of VDR was negatively correlated with SOFA score and lactate and C-reactive protein levels [25,26]. The mechanism through which vitamin D deficiency affects sepsis may be considered as a hypothesis, whereby a decrease in VDRs reduces the activity of the immune response, which leads to a decrease in the anti-inflammatory reaction required for sepsis. In a pneumococcal pneumonia animal model, vitamin D was shown to be involved in innate immunity; vitamin D upregulates toll-like receptor 2 (TLR2) and nucleotide-binding oligomerization domain 2 (NOD2) and induces antibacterial human neutrophil peptides and LL-37 that are involved in adaptive immunity. Vitamin D supplementation reduces the induction of the suppressors of cytokine signaling (SOCS) [27].

Currently, no studies have reported on whether the level of vitamin D deficiency affects the immune system, although vitamin D deficiency has been reported to affect the severity of infection [28]. It is also not known whether vitamin D activates the immune system when a certain cut-off value or more is exceeded or whether it is activated quantitively according to the vitamin D concentration. In a previous randomized controlled trial, vitamin supplementation was not effective in improving mortality rates in sepsis patients with vitamin D insufficiency when the intensity of the vitamin D levels was not considered [29]. In this study, severe vitamin D deficiency, not overall vitamin D deficiency, affected sepsis mortality, suggesting that the effect of vitamin D supplementation on the mortality of sepsis patients with severe vitamin D deficiency may need to be re-evaluated. The results of this study may consider the possibility that the analysis did not produce statistically significant results, as the vitamin D deficiency group accounted for the majority of the total population. Further studies may be needed that use larger study populations.

Severe vitamin D deficiency increased 14-day, 28-day, and in-hospital mortality, yet it did not affect 7-day mortality. In sepsis, the immune system’s effects in early death (before day 5) and late death (on or after day 5) may be different, although the effect of neutrophils in early death and lymphocytes in late death may be greater [30]. This suggests that the effect of vitamin D on the immune system may be greater on adaptive immunity than on innate immunity. In-hospital morbidity may also be related to the prolonged immune suppression in the later stage of sepsis, and vitamin D deficiency along with the comorbidity disease may result in reduced immune responses [31,32]. Vitamin D insufficiency was identified in 92% of sepsis patients in this study, suggesting that checking vitamin D levels may be necessary in patients hospitalized with sepsis. The risk group for sepsis may be similar to the risk group for vitamin D deficiency; thus, checking vitamin D levels may be needed to prevent fractures or osteoporosis, which are related to bone metabolism in addition to the effects on the immune system. As previous studies have shown, SOFA and the Charlson comorbidity index can affect the increased mortality from sepsis [5].

The limitations of this study are as follows. Firstly, the main limitation is the possibility of reverse causality. Since this study is not an interventional study, the possibility should be considered that vitamin D deficiency is a result rather than a cause of sepsis; this should be evaluated by further interventional study. Secondly, this study confirmed that severe vitamin D deficiency increased sepsis mortality; however, no effect was shown for vitamin D deficiency. This may be due to the high rate of vitamin D deficiency in the overall population, which means that deficiency levels are currently insufficient to achieve statistically significant results. Next, this study did not measure vitamin D receptors; vitamin D receptors may directly explain the mechanism related to vitamin D deficiency in mortalities from sepsis. Finally, this study was a prospective cohort study, so no intervention was applied. Further studies may be needed to identify whether vitamin D supplementation improves the sepsis mortality rate.

In conclusion, 74.4% of sepsis patients suffered from vitamin D deficiency. Moreover, severe vitamin D deficiency affected the increase in the 14-day, 28-day, and in-hospital mortality rates, meaning that checking vitamin D levels is required in patients hospitalized with sepsis. In sepsis patients with vitamin D deficiency, further studies are needed regarding the short- and long-term prognosis of sepsis through vitamin D supplementation.

## Figures and Tables

**Figure 1 nutrients-15-04309-f001:**
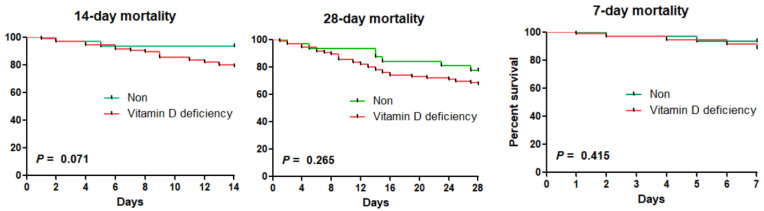
Kaplan–Meier curves for mortalities according to vitamin D deficiency.

**Figure 2 nutrients-15-04309-f002:**
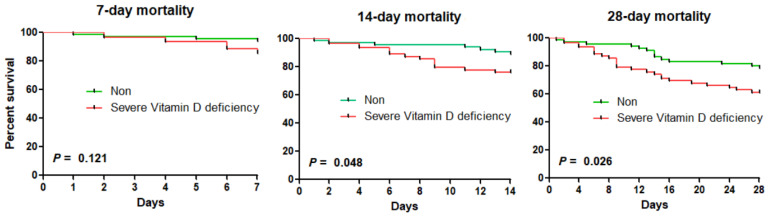
Kaplan–Meier curves for mortalities according to severe vitamin D deficiency.

**Table 1 nutrients-15-04309-t001:** Baseline characteristics of the study population.

	Total (n = 129)	Vitamin D Insufficiency (n = 119)	Vitamin D Deficiency (n = 96)	Severe Vitamin D Deficiency (n = 62)
Age, years	74 ± 13 (66–83)	74 ± 13 (66–83)	75 ± 13 (68–83)	74 ± 15 (66–83)
Male	60 (46.5)	54 (45.4)	42 (43.8)	27 (4.35)
Vitamin D level (ng/mL)	14.5 (6.2–20.8)	11.1 (6.0–17.8)	8.5 (5.0–14.1)	6.1 (4.6–8.4)
Charlson comorbidity index	5 (3–6)	5 (3–6)	5 (3–6)	5 (3–6)
Grade of infection				
High-grade	83 (64.3)	76 (63.9)	60 (62.5)	40 (64.5)
Low-grade	46 (35.7)	43 (36.1)	36 (37.5)	22 (36.1)
SOFA score	7 (5–10)	7 (5–10)	7 (5–10)	7 (5–9)
Septic shock	60 (46.5)	57 (47.9)	6 (47.9)	29 (46.8)
Bacteremia	45 (34.9)	41 (34.5)	36 (37.5)	21 (33.9)
ICU admission	78 (60.5)	75 (63.0)	63 (65.6)	39 (62.9)
CRP (mg/dL) *	10.5 (4.2–21.5)	11.2 (4.4–22.2)	10.2 (4.4–22.1)	11.2 (4.5–20.4)
Procalcitonin (ng/mL) *	1.0 (0.3–7.1)	1.15 (0.28–10.51)	1.08 (0.27–7.02)	0.90 (0.27–5.39)
Initial lactate (mmol/L) *	2.4 (1.7–5.9)	2.5 (1.7–6.0)	2.7 (1.8–6.1)	2.3 (1.7–5.8)
Time to antibiotics administration, minutes	119 (72–240)	119 (72–230)	115 (72–213)	130 (72–220)
Appropriateness of empirical antibiotics administration	99 (76.7)	92 (77.3)	72 (75.0)	50 (80.6)
All-cause 7-day mortality	13 (10.1)	13 (10.9)	11 (11.5)	9 (14.8)
All-cause 14-day mortality	22 (17.1)	22 (18.5)	20 (20.8)	15 (24.6)
All-cause 28-day mortality	38 (29.5)	36 (30.3)	31 (32.3)	24 (38.7)
In-hospital mortality	36 (27.9)	34 (28.6)	30 (31.3)	22 (35.5)

NOTE. Data are expressed as number (%) of patients or median (interquartile range) unless otherwise indicated. SOFA: sequential organ failure assessment; ICU: intensive care unit; CRP: C-reactive protein. * Reference range: CRP 0–0.5 mg/dL; Procalcitonin < 0.05 ng/mL; Lactate 0.5–1.6 mmol/L.

**Table 2 nutrients-15-04309-t002:** Risk factors for outcomes in sepsis patients with vitamin D deficiency.

	7-Day Mortality		14-Day Mortality		28-Day Mortality		In-Hospital Mortality	
	aHR (95% CI)	*p*-Value	aHR (95% CI)	*p*-Value	aHR (95% CI)	*p*-Value	aHR (95% CI)	*p*-Value
Vitamin D deficiency	1.78 (0.39–8.08)	0.454	3.20 (0.74–13.81)	0.120	1.51 (0.66–3.45)	0.332	1.63 (0.66–4.01)	0.287
Age	1.02 (0.97–1.07)	0.487	0.98 (0.95–1.02)	0.361	0.99 (0.97–1.02)	0.517	0.98 (0.95–1.01)	0.182
CCI	1.10 (0.85–1.42)	0.479	1.30 (1.11–1.53)	0.002	1.22 (1.07–1.39)	0.003	1.31 (1.14–1.50)	<0.001
SOFA	1.22 (1.02–1.44)	0.025	1.08 (0.96–1.22)	0.203	1.11 (1.01–1.22)	0.027	1.10 (0.99–1.22)	0.060
Shock	1.33 (0.66–2.69)	0.430	1.38 (0.85–2.24)	0.198	1.25 (0.83–1.90)	0.285	1.17 (0.75–1.83)	0.483

aHR: adjusted hazard ratio; CI: confidence interval; CCI: Charlson comorbidity index; SOFA: sequential organ failure assessment.

**Table 3 nutrients-15-04309-t003:** Risk factors for outcomes in sepsis patients with severe vitamin D deficiency.

	7-Day Mortality		14-Day Mortality		28-Day Mortality		In-Hospital Mortality	
	aHR (95% CI)	*p*-Value	aHR (95% CI)	*p*-Value	aHR (95% CI)	*p*-Value	aHR (95% CI)	*p*-Value
Severe vitamin D deficiency	2.80 (0.85–9.23)	0.090	2.57 (1.03–6.43)	0.043	2.28 (1.17–4.45)	0.016	2.11 (1.02–4.36)	0.044
Age	1.02 (0.97–1.07)	0.496	0.98 (0.95–1.02)	0.336	0.99 (0.96–1.02)	0.486	0.98 (0.96–1.01)	0.244
CCI	1.10 (0.85–1.43)	0.467	1.31 (1.12–1.54)	0.001	1.23 (1.08–1.41)	0.002	1.30 (1.13–1.49)	<0.001
SOFA	1.24 (1.04–1.48)	0.016	1.12 (0.99–1.27)	0.077	1.13 (1.03–1.24)	0.010	1.12 (1.01–1.24)	0.034
Shock	1.31 (0.67–2.56)	0.437	1.33 (0.81–2.20)	0.267	1.21 (0.80–1.82)	0.365	1.10 (0.70–1.74)	0.676

aHR: adjusted hazard ratio; CI: confidence interval; CCI: Charlson comorbidity index; SOFA: sequential organ failure assessment.

## Data Availability

All data used in analysis of this manuscript are freely available by contacting the corresponding author.

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
