# Peer review of "Effects of Vitamin D Deficiency on Sepsis"

_nutrients, 2023, doi:10.3390/nu15204309_

Round 1
Reviewer 1 Report
Thank you for the opportunity to review this manuscript. This study was conducted to evaluate the relationship between vitamin D deficiency and mortality in sepsis patients. I have only some suggestions to improve the manuscript.
In the introduction I think that authors could be better explore the functions of vitamin D in immune response, deepening the role of vitamin D in immune response. Vitamin D can modulate the innate and adaptive immune response, reduce T helper cell differentiation and proliferation and induce a more tolerogenic immune response than a pro infiammatory status. In addition, I suggest to implement the references.
Methods, results and discussione are well structured.
Minor editing of English language required
Author Response
1. In the introduction I think that authors could be better explore the functions of vitamin D in immune response, deepening the role of vitamin D in immune response. Vitamin D can modulate the innate and adaptive immune response, reduce T helper cell differentiation and proliferation and induce a more tolerogenic immune response than a pro infiammatory status. In addition, I suggest to implement the references.
→ Thank you for your valuable comments. We revised the introduction according to the reviewer’s recommendation.
2. Methods, results and discussione are well structured
→ We appreciate your kind review.
Reviewer 2 Report
Authors of paper “Effects of Vitamin D Deficiency on Sepsis” showed an increased mortality rate in patients with sepsis, who had severe vitamin D deficiency. The article presents the results of prospective cohort study, carried out in a group of 129 patients, after introduction of the Sepsis-3 definition.
The effect of vitamin D deficiency on sepsis is controversial. Previous prospective studies and meta-analyses have reported both - association of vitamin D deficiency with increased mortalities in sepsis and lack of such relationship.
1. The main limitation of the study is related to phenomenon of “reverse causality”.
In inflammatory diseases low levels of 25OHD are commonly observed. However nature of this asscociation is not clear. Hypovitaminosis D might be the both – the cause or the consequence (and an indicator) of inflammatory diseases, since 25OHD was shown to be an acute phase reactant . In such settings, caution must be applied to interpretation of the above data although reverse causality cannot be excluded. Phenomenon of “reverse causality” applies to a situation where an outcome seemingly determines the cause, thus leading to an erroneous conclusion where in fact an opposite relationship might exist. In this case, increased mortality in sepsis might have been attributed to severe vitamin D deficiency 25OHD concentrations, while a more severe course of sepsis (due to as-yet unidentified factors) might have resulted in lower 25OHD levels, as the consequence of a decline in 25OHD typical for any acute illness. On the basis of presented data it is not possible to conclude what is a reason and what is a consequence, since the 25OHD was measured during sepsis, not before.
2. Line 17-19 and - conclusion should be changed, because it does not result from this research, which was not intervention study
3. Table 1 – I suggest to add reference range for CRP, lactate and procalcitonin
4. Methods – Measurments of vitamin D
a. Lack of information about intra-assay and inter-assay variation
b. when was 25OHD determined? On admission or later? How many measurments of 25 OHD were carried out for every patient?
5. Line 212 – It is not true [Wang, Z.; Joshi, A.; Leopold, K.; Jackson, S.; Christensen, S.; Nayfeh, T.; Mohammed, K.; Creo, A.; Tebben, P.; Kumar, S. Association of vitamin D deficiency with COVID-19 infection severity: Systematic review and meta-analysis. Clin. Endocrinol.2022, 96, 281–287 ]
6. Have Authors observed any influence of season ( Winter vs. Summer) on vitamin D levels?
7. What were other determinants of 25OHD levels? BMD, age?
I have no comments.
